# The transcriptional repressor Blimp1/PRDM1 regulates the maternal decidual response in mice

Mubeen Goolam [1,2], Maria-Eleni Xypolita[1], Ita Costello[1], John P. Lydon[3], Francesco J. DeMayo[4], Elizabeth K. Bikoff[1], Elizabeth J. Robertson [1✉] & Arne W. Mould [1]

The transcriptional repressor Blimp1 controls cell fate decisions in the developing embryo and adult tissues. Here we describe Blimp1 expression and functional requirements within maternal uterine tissues during pregnancy. Expression is robustly up-regulated at early post-implantation stages in the primary decidual zone (PDZ) surrounding the embryo. Conditional inactivation results in defective formation of the PDZ barrier and abnormal trophectoderm invasion. RNA-Seq analysis demonstrates down-regulated expression of genes involved in cell adhesion and markers of decidualisation. In contrast, genes controlling immune responses including IFNγ are up-regulated. ChIP-Seq experiments identify candidate targets unique to the decidua as well as those shared across diverse cell types including a highly conserved peak at the Csf-1 gene promoter. Interestingly Blimp1 inactivation results in up-regulated Csf1 expression and macrophage recruitment into maternal decidual tissues. These results identify Blimp1 as a critical regulator of tissue remodelling and maternal tolerance during early stages of pregnancy.

[1] Sir William Dunn School of Pathology, University of Oxford, Oxford OX1 3RE, UK. [2] Department of Human Biology, University of Cape Town, Cape Town 7925, South Africa. [3] Molecular and Cellular Biology, Baylor College of Medicine, Houston 77030, USA. [4] Reproductive and Developmental Biology Laboratory, NIEHS, Research Triangle Park, NC 27709, USA. ✉email: elizabeth.robertson@path.ox.ac.uk

Divergent patterns of gene expression underlie the establishment of cell identity. For the most part, this reflects the induction of gene expression by transcriptional activators. However, in some cases cell fate decisions, especially those made within an already committed lineage, may be due to selective silencing of an ongoing transcriptional programme that is associated with a developmental switch controlled by a cell-type specific repressive complex. Much has been learned over recent years about the constitutive epigenetic machinery responsible for gene silencing—including the activities of key enzymatic components such as G9a, Lsd-1 and Hdac family members. However, relatively little is known about cell type specific repressors guiding target site selection.

We have been studying the zinc finger SET domain protein Blimp1 encoded by the *Prdm1* gene —originally cloned as a post-inductive repressor of type I IFNβ gene expression[1]- and subsequently identified as the key transcription factor controlling terminal plasma cell differentiation[2]. Its functional role in the B cell lineage has been extensively characterised. Its ability to silence expression of well-described target genes including c-Myc, Pax5, Bcl6 and CIITA, and consequently cause termination of B cell identity in favour of terminally differentiated plasma cell functionality has been intensely investigated[3]. Similarly in the context of the early embryo, Blimp1 silences the default somatic programme allowing a small subset of primordial germ cell (PGC) progenitors to avoid responsiveness to BMP/Smad signals and become committed to acquire a germ cell fate[4,5].

Loss of function Blimp1 mutant embryos arrest at around embryonic day (E) 10.5 due to defective placental morphogenesis[5,6]. Blimp1 expression is essential for specification of a distinct sub-set of trophoblast giant cells, the SpA-TGC, that migrate into the uterine maternal tissue to surround, invade and remodel the maternal blood vessels[6,7]. Interestingly, our recent sc-RNA-Seq analysis of specialized cell types at the fetal-maternal interface at mid-gestation stages identified a discrete sub-population of maternal Blimp1+ cells co-expressing high levels of the decidual stromal marker Prl8a2[7].

In the present study we perform immunostaining experiments to further investigate Blimp1 expression within the maternal uterine environment. Expression is robustly upregulated at early post-implantation stages in the primary decidual zone (PDZ) surrounding the embryo. To explore Blimp1 functional contributions we exploit the well-characterised progesterone receptor Cre (PR-Cre) strain[8] to selectively eliminate Blimp1 expression in the maternal uterine environment. The loss of function mutation compromises the decidualisation response and results in loss of PDZ barrier formation, ectopic trophoblast expansion, increased macrophage invasion and ultimately, embryonic lethality.

## Results

**Upregulated Blimp1 expression during implantation**. In the virgin uterus, Blimp1 expression is restricted to a few scattered cells within the stroma (Fig. 1a). A discrete population of Blimp1+ stromal cells immediately adjacent to the uterine luminal epithelium (LE) was readily detectable at embryonic day (E3.5) of pregnancy, prior to embryo implantation. Coincident with embryo attachment within the uterine crypts 24 hours (hr) later we observe a marked increase in Blimp1 expression in the uterine LE immediately adjacent to the trophectoderm. Blimp1 expression is strongly upregulated during formation of the PDZ surrounding the embryo. Expression persists in the PDZ at E5.5 and E6.5. Immunostaining results were confirmed by Western blot analysis (Fig. 1b).

**Blimp1 inactivation compromises maternal decidual response**. To investigate Blimp1 functional contributions, we exploited the PR-Cre deleter strain, proven to be a valuable tool for studying gene function in the uterine LE and stroma during pregnancy[8]. To generate females lacking Blimp1 function in progesterone responsive tissues, hereinafter referred to as Blimp1 mutants, we crossed PR-Cre males carrying a Blimp1 null allele[5] with females homozygous for the Blimp1 conditional allele[9]. Loss of Blimp1 expression in maternal decidual tissues was confirmed by immunostaining at E6.5 (Fig. 1c).

We observed normal numbers of decidual swellings in Blimp1 mutant females at E6.5. (Fig. 2a). The average number of decidua/mouse ± standard error of the mean (SEM) was 9.88 ± 0.42 for wild types ($n = 25$) and 9.38 ± 0.38 for mutants ($n = 24$). These results strongly suggest that embryo attachment and implantation proceed normally. However, as judged by reduced levels of Chicago blue uptake (Fig. 2b) mutant deciduae are significantly smaller in comparison to wild type. Alkaline phosphatase staining, a marker of stromal decidualisation was significantly reduced (Fig. 2c, Supplementary Fig. 1a). As assessed by BrdU incorporation and Ki67 staining these differences cannot simply be explained due to reduced proliferative capacity (Supplementary Fig. 1b, 1c). Collectively, these results demonstrate that loss of Blimp1 function compromises the maternal decidual response.

Embryos normally become confined to the anti-mesometrial region of the decidua shortly after implantation. In contrast, Blimp1 mutant embryos were often located in more mesometrial regions of the uterine crypts (Fig. 2d, Supplementary Fig. 1d). In Blimp1 mutant females at E6.5, both the embryos and surrounding PDZ display morphological disturbances. Additionally, the cell density of the PDZ is significantly reduced (Fig. 2e).

**Loss of PDZ barrier and abnormal trophectoderm invasion**. To identify the extra-embryonic ectoderm (ExE) and trophoblast cell populations at early post-implantation stages, next we examined expression of the Tbox transcription factor Eomesodermin (Eomes) (Fig. 3a). We observed extensive invasion of Eomes+ trophectoderm giant cells into the maternal stroma in Blimp1 mutants. Whole mount staining of dissected embryos at E5.5 likewise shows that they retain the Oct4+ epiblast and overlying Eomes+ visceral endoderm populations. However, the morphology of extra-embryonic tissues is highly disturbed. Thus, the ExE which normally provides important trophic signals to the underlying epiblast[10,11] is selectively reduced.

To better visualise the behaviour of the ExE and embryonic cell populations, we made use of a paternally inherited ubiquitously expressed Rosa26-membraneTomato knock-in allele[12]. Confocal imaging of thick sections through E5.5 implantation sites demonstrate extensive ectopic growth of the mTomato+-labelled embryonic cells into the Blimp1 mutant maternal decidual tissues. As judged by cell size and keratin-8 immuno-reactivity, these invasive cells mostly represent primary trophoblast giant cells (Fig. 3c). We failed to recover viable intact embryos beyond E6.5.

Formation of the avascular PDZ immediately surrounding the embryo is associated with robust expression of the tight junction protein ZO-1 (Fig. 3d). Consistent with decreased tight junction density, ZO-1 staining is markedly reduced around the mutant E5.5 implantation sites (Fig. 3d, e). Moreover, transmission electron microscopy (TEM) analysis revealed a highly disturbed cellular architecture at E6.5 (Fig. 3f). In contrast to the wild-type PDZ, characterised by densely packed cells and extensive tight junction formation, the mutant decidua cells are very loosely packed and there were only a few sporadic tight junctions. Interestingly, this cellular morphology closely resembles that of pre-decidual stromal cells[13]. Collectively, these experiments demonstrate that Blimp1 function is required in the maternal

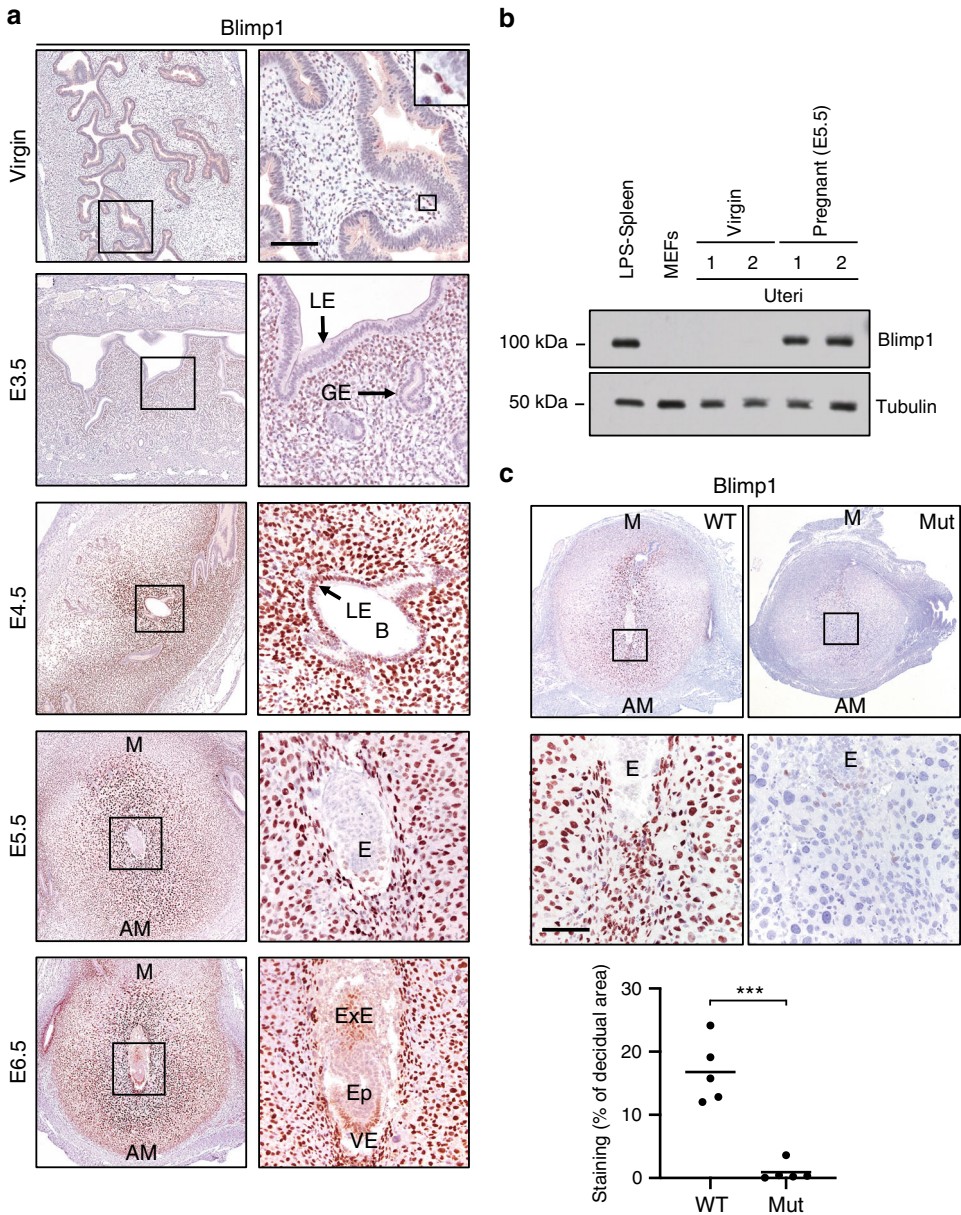

**Fig. 1 Induction of Blimp1 protein expression during pregnancy and PR-Cre-driven deletion in decidual tissues. a** Blimp1 IHC in virgin, E3.5, E4.5, E5.5 and E6.5 wild-type uteri. In virgins, Blimp1 was detected in a small number of cells underlying the luminal epithelium. Upon fertilisation, Blimp1 was induced in uterine stroma (E3.5) followed by strong upregulation at the site of implantation in both the decidualising stroma and luminal epithelium (E4.5-E6.5). Results are representative of triplicate staining experiments performed using independent samples. **b** Western blot analysis confirmed strong induction of Blimp1 protein in the uterus during pregnancy. Source data are provided as a Source Data file. Duplicate experiments were performed with comparable results. **c** IHC confirms loss of Blimp1 protein in decidual stromal cells in Blimp1 mutant mice at E6.5. Data from 5 tissue sections from triplicate decidual samples per genotype. Source data are provided as a Source Data file. Bars represent mean. Two-tailed unpaired Student's $t$-test ***$p = 1.36 \times 10^{-4}$. LE = luminal epithelium, GE = glandular epithelium, B = blastocyst, E = embryo, ExE = extra-embryonic ectoderm, Ep = epiblast, VE = visceral endoderm, M = mesometrial, AM = antimesometrial. Scale bar = 100 μm.

tissue to establish the densely packed PDZ that normally acts as a barrier to constrain TE invasion.

**RNA-Seq identifies gene expression changes.** To further characterise cellular defects in Blimp1 mutant decidua, we performed transcriptional profiling experiments. RNA-Seq analysis of wild-type and mutant-decidual tissues demonstrates gene expression changes detectable at E5.5 (Supplementary Data 1). More pronounced differences became evident by E6.5 (Fig. 4a, Supplementary Data 2). Hierarchical clustering shows that E6.5 Blimp1 mutant transcripts are more similar to E5.5 wild-type deciduae

than they are to E6.5 wild-type deciduae (Supplementary Fig. 2a). Based on statistical significance (DESeq2 FDR < 0.05), greater than 2-fold changes in expression, and an expression level confidence filter (FPKM ≥ 1 in all samples of either genotype), 703 genes were upregulated and 458 genes were found to be downregulated in E6.5 Blimp1 mutant decidua in comparison to wild type. Gene ontology (GO) analysis of upregulated genes shows a significant enrichment for genes associated with responses to external biotic stimulus, regulation of the immune response, regulation of the defence response, leukocyte activation, and cell chemotaxis (Fig. 4b, Supplementary Data 3). Recent experiments demonstrate that many of

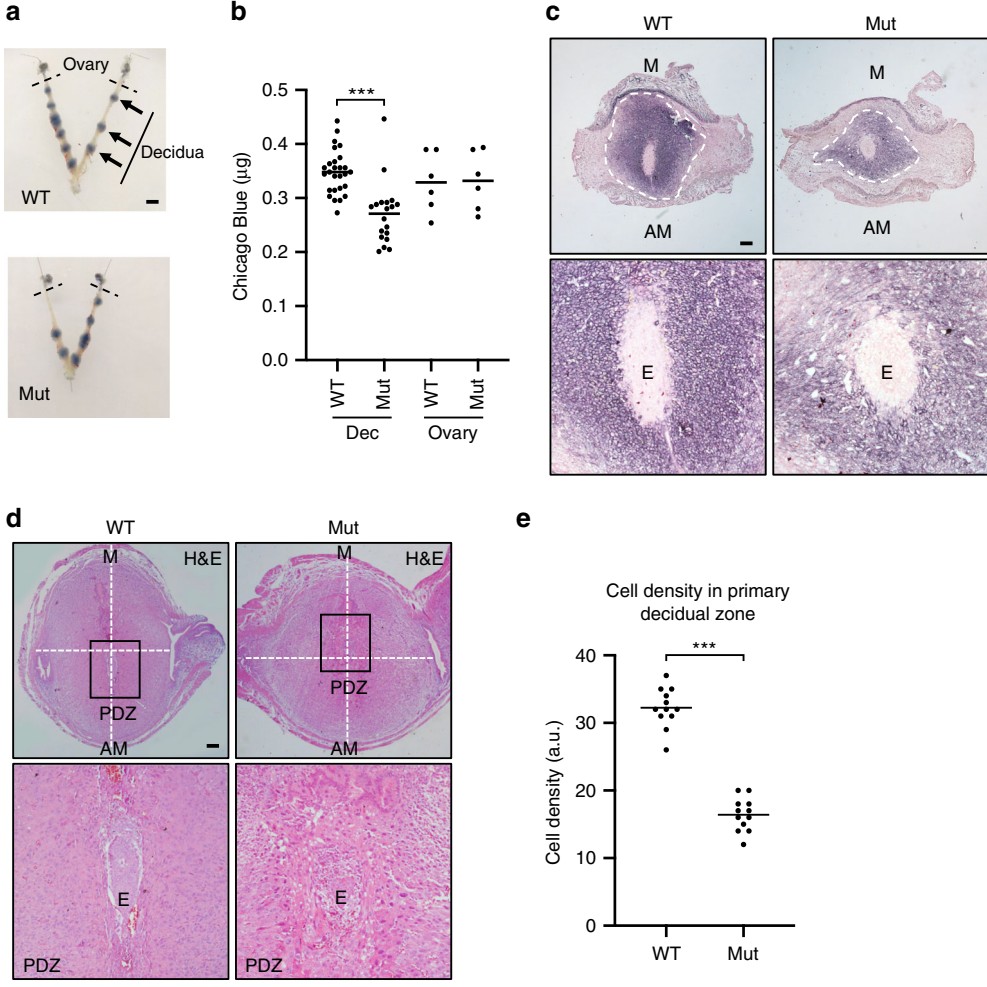

**Fig. 2 Blimp1 mutants are receptive to implantation but display gross decidual and embryonic defects. a** Sites of decidualisation at E6.5 appear grossly similar between wild-type and Blimp1 mutant uteri, as visualised by Chicago Blue dye injection ($n = 3$ per group). **b** Chicago Blue content is reduced in decidua but not the ovaries of mutants ($0.271 \pm 0.013\ \mu g$/decidua) in comparison to wild type ($0.348 \pm 0.008\ \mu g$/decidua) mice. Data from 3 Blimp1 mutant mice (19 deciduae and 6 ovaries) and 3 wild-type mice (27 deciduae and 6 ovaries) are shown. Bars represent mean. Two-tailed unpaired Student's $t$-test ***$p = 3.49 \times 10^{-6}$. Source data are provided as a Source Data file. **c** Alkaline phosphatase staining of wild-type and Blimp1 mutant E5.5 decidua shows a reduction in intensity and the spread of alkaline phosphatase staining in Blimp1 mutant decidua indicative of impaired decidualisation ($n = 3$ decidua per genotype). **d** H&E staining of E6.5 deciduae. Embryos in Blimp1 mutant mice are highly abnormal and display mis-localised implantation ($n = 4$ decidua per genotype). **e** Cell density analysis of DAPI-stained nuclei identifies reduced cell density in the PDZ of Blimp1 mutants. Data from 12 tissue sections from four samples per genotype are shown. Bars represent mean. Two-tailed unpaired Student's $t$-test ***$p = 1.08 \times 10^{-12}$. Source data are provided as a Source Data file. PDZ = primary decidual zone, M = mesometrial, AM = antimesometrial, E = embryo. Scale bars = **a** 4 mm or **c–d** 100 μm.

these genes are normally silenced in decidual tissues to maintain an immunologically privileged environment and protect the developing embryo[14]. In contrast, the 458 downregulated genes were enriched for genes involved in negative regulation of peptidase activity, cell adhesion, regulation of body fluid levels, female pregnancy and reproductive structure development (Fig. 4b, Supplementary Data 3). For example, expression of *Prl* family members selectively (*Prl8a2*) or predominantly (*Prl3c1* and *Prl6a1*) expressed by decidual cells at early stages of pregnancy[15] is downregulated in Blimp1 mutants (Fig. 4c), strengthening the evidence suggesting decidualisation is compromised.

The steroid hormones progesterone and oestrogen are known to play essential roles controlling gene expression during stromal cell decidualisation[16]. When we used gene set enrichment analysis to compare differentially expressed genes in E6.5 Blimp1 mutants with predicted progesterone and oestrogen responsive genes (based on proximal PR and Esr1 binding) we observed a highly significant correlation (FDR $q$-value > 0.0001) between downregulated genes in Blimp1 mutant decidua and progesterone receptor target genes

(Fig. 4d). Similar but less pronounced correlations were detectable for oestrogen receptor target genes (FDR $q$-value = 0.024). These results strongly suggest that pregnancy hormonal-driven decidualisation is impaired in Blimp1 mutants.

Consistent with results above that demonstrate increased trophoblast spreading and invasion in mutant decidua (Fig. 3), expression of parietal trophoblast giant cell (P-TGC)-restricted *Prl3d1, 3d2 and 3d3* (and *Prl7a1* at this stage of gestation)[15] is markedly increased. Additionally, the metallopeptidase (MMP)-inhibitor *Timp3*, selectively expressed within the PDZ immediately surrounding the embryo[17], is downregulated (Fig. 4c). Blimp1 mutant E6.5 decidua display increased *Mmp11* (Supplementary Data 2). It seems likely that increased MMP activity observed in Blimp1 mutant decidua facilitates trophoblast spreading.

It is well known that Blimp1 mediated repression silences interferon signalling and expression of multiple interferon-responsive genes[1,18–20]. Interestingly, here we observe upregulated expression of multiple IFNγ-inducible genes (Fig. 4e,

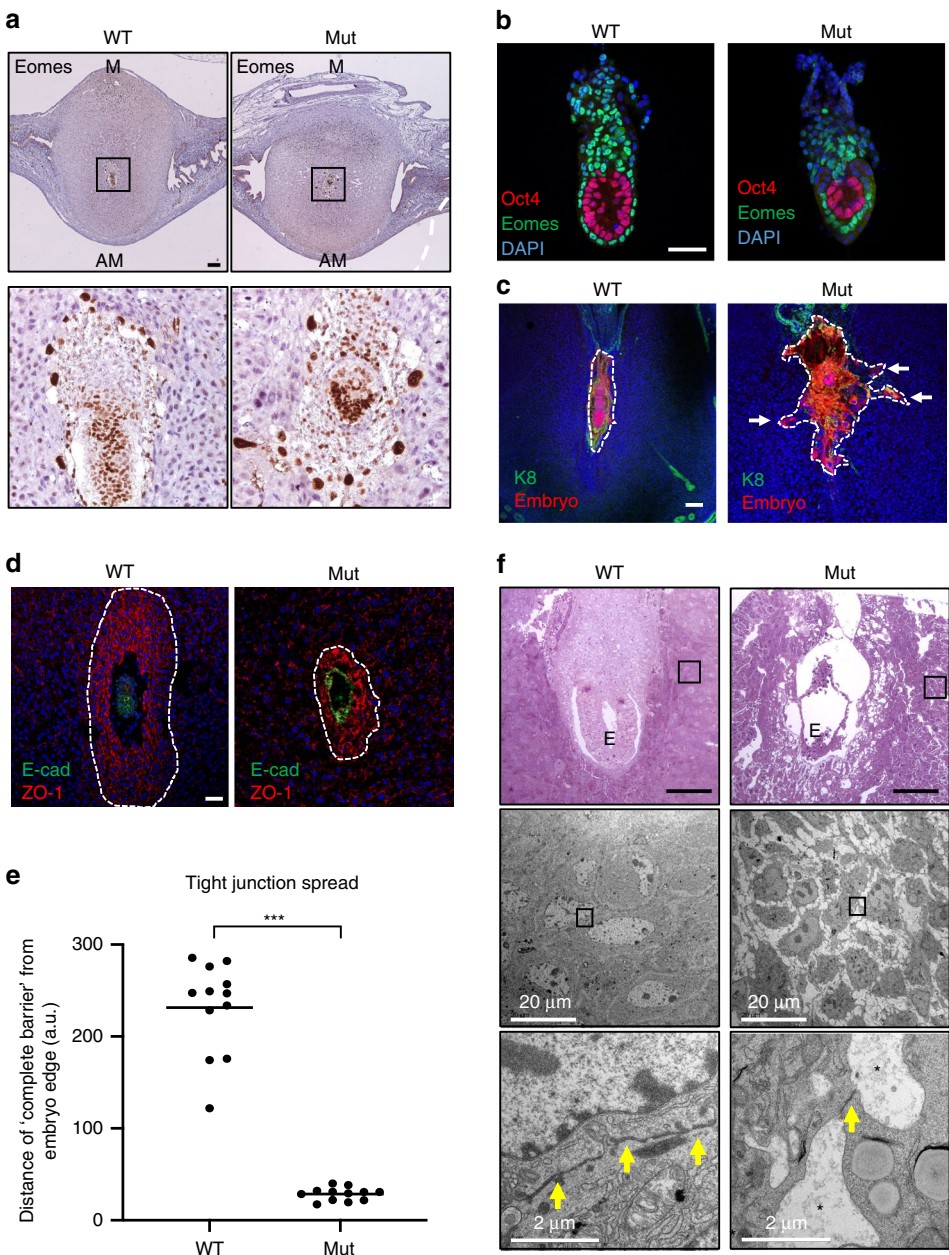

**Fig. 3 Loss of the primary decidual zone barrier in Blimp1 mutants. a** IHC identifies invasion of Eomes positive trophoblasts into the PDZ in Blimp1 mutant decidua ($n = 5$ decidua per genotype). **b** IF staining of Oct4 and Eomes in isolated E5.5 embryos from wild-type and Blimp1 mutant mice counterstained with DAPI. Embryos isolated from Blimp1 mutant deciduae display normal epiblast morphology but distinct disruption of the extraembryonic ectoderm ($n = 5$ embryos per genotype). **c** Immunofluorescence staining of RFP (to identify embryonic cells) and K8 (trophoblasts) in wild type; mTmG and Blimp1 mutant; mTmG decidua (E5.5) counterstained with DAPI. White dotted lines outline embryos. White arrows in Blimp1 mutant;mTmG images indicate migrating trophoblast cells. RFP-labelled embryos from Blimp1 mutant; mTmG mice are highly disrupted and invade into the PDZ ($n = 5$ per genotype). **d** IF staining of E-cad (embryo) and ZO-1 in E5.5 mutant and wild-type decidua counterstained with DAPI. Reduced ZO-1 staining indicates impaired tight junction formation in the PDZ of mutants ($n = 5$ per genotype). White dotted lines outline the PDZ. **e** Reduced ZO-1 positive area forming a 'complete barrier' in the PDZ of E5.5 mutants demonstrates impaired barrier formation. Data from 12 tissue sections from four samples per genotype are shown. Bars represent mean. Two-tailed unpaired Student's t-test ***$p = 2.01 \times 10^{-12}$. Source data are provided as a Source Data file. **f** TEM of E6.5 PDZ confirms loss of a complete barrier in mutants ($n = 3$ per genotype). Top panel shows sections of resin embedded samples counterstained with toluidine blue for general morphology before TEM was undertaken. Yellow arrows indicate tight junctions at cell–cell borders. Asterisks mark intercellular spaces. E = embryo, M = mesometrial, AM = antimesometrial, a.u. = arbitrary units. Scale bars = 100 μm unless otherwise indicated.

Supplementary Data 4) as well as increased levels of *Ifnγ* transcripts (Fig. 4f) in mutant decidua. Upregulated gene categories related to leukocyte activation and cell chemotaxis were also over-represented. Moreover, well-known macrophage marker genes including *Itgam* and *Adgre1*, encoding Cd11b and

F4/80 respectively, were also found to be strongly upregulated (Supplementary Fig. 2b).

**ChIP-Seq analysis identifies candidate Blimp1 target genes.** To further investigate Blimp1 functional contributions during

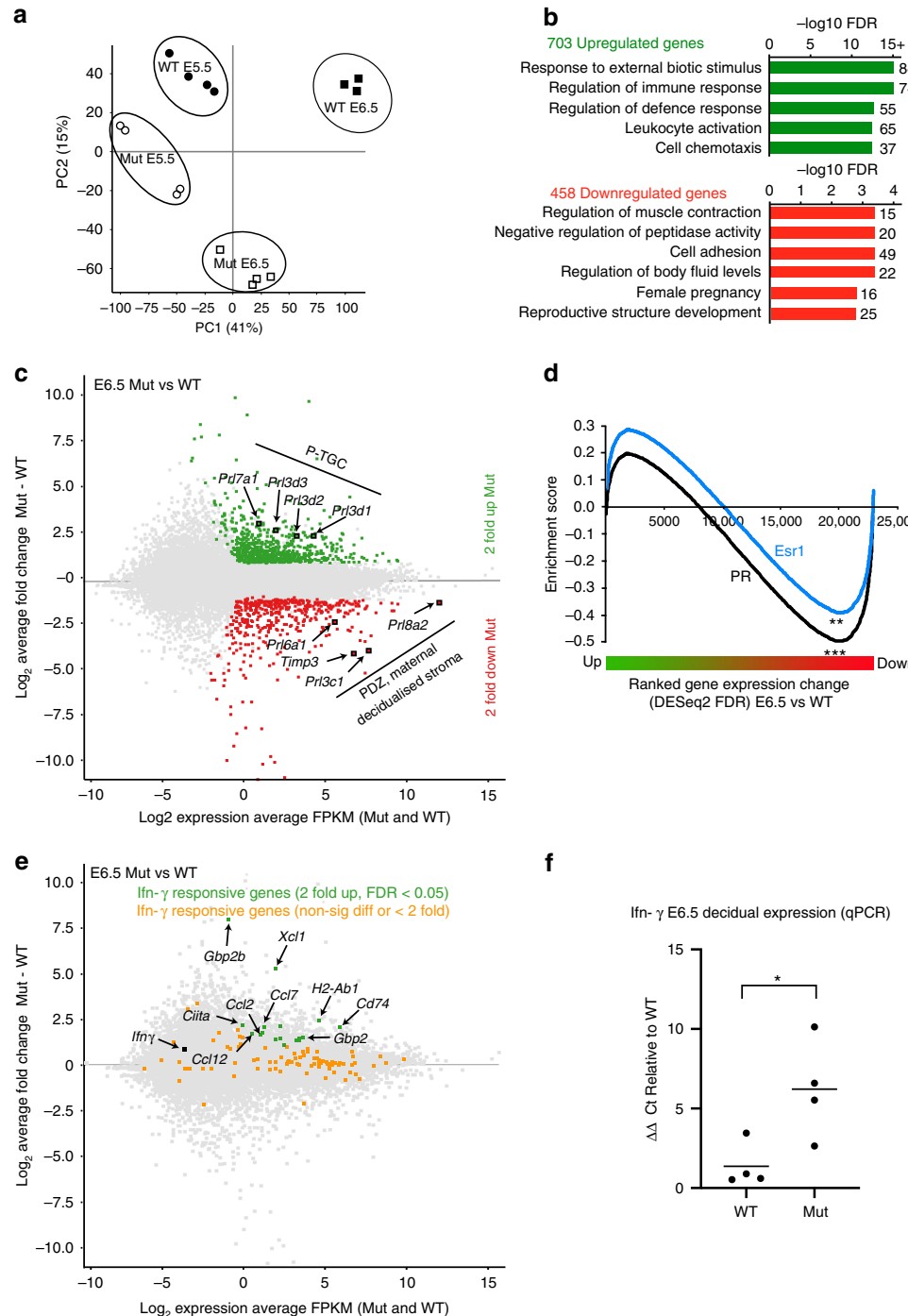

**Fig. 4 Transcriptional changes in E5.5 and E6.5 Blimp1 mutant decidua. a** RNA-Seq analysis of Blimp1 mutant and wild-type decidua at E5.5 and E6.5. Principal component analysis based on the expression of all genes ($n = 34,245$) clearly separates samples according to stage and Blimp1 genotype. The most pronounced transcriptional differences between the genotypes are observed at E6.5. **b** Over-represented gene categories among ≥2-fold up-($n =$ 703) and down-($n = 458$) regulated genes in E6.5 mutants based on GO Biological process with affinity propagation as analysed using Webgestalt 2019 with Benjamini-Hochberg multiple testing correction. **c** Differential expression of *Prl* genes and *Timp3* in E6.5 mutants. *Prl* genes that are selectively (*Prl8a2*) or predominantly expressed (*Prl3c1* and *Prl6a1*) in decidual cells at early stages of gestation are decreased. In contrast, P-TGC restricted *Prl genes* (*Prl3d1, 3d2 and 3d3*, and *Prl7a1* at this stage of gestation) are upregulated. The PDZ-associated *Timp3* gene is downregulated. **d** GSEA comparing DESeq2 FDR-ranked genes in E6.5 mutants with uterine progesterone and oestrogen target genes[58,59]. Downregulated gene expression in E6.5 Blimp1 mutants significantly correlates with both progesterone- and oestrogen-target genes. **FDR $p = 0.024$. ***FDR $p = 0.000$. **e** Multiple IFNγ-inducible genes are upregulated in E6.5 Blimp1 mutant decidua. *Ifnγ* is increased in Blimp1 mutant decidua but due to low level of expression was excluded from RNA-Seq analysis based on the FPKM ≥ 1 minimum level filter. **f** QPCR analysis confirms significant upregulation of *Ifnγ* transcripts in E6.5 Blimp1 mutant decidua. Data from four samples per genotype are shown. Bars represent mean. Two-tailed unpaired Student's *t*-test *$p = 0.0289$. Source data are provided as a Source Data file.

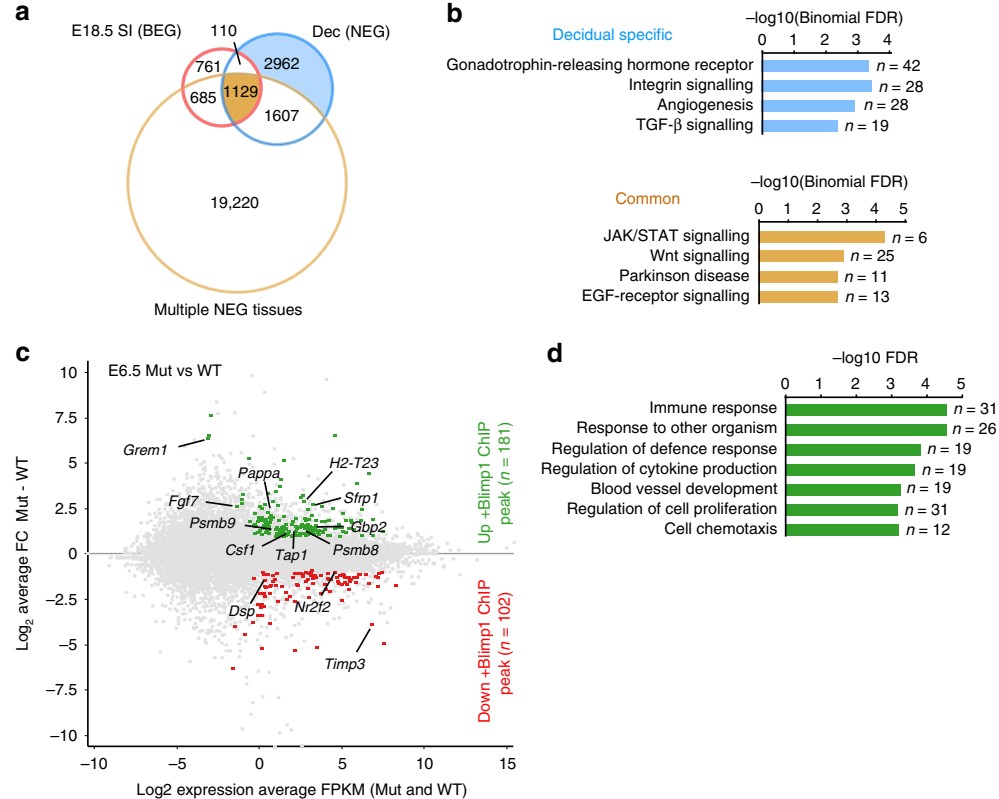

**Fig. 5 ChIP-Seq analysis of E6.5 decidua identifies potential Blimp1 target genes. a** Venn diagram overlaps of genome-wide Blimp1 binding sites identified by ChIP-Seq analysis of; E6.5 NEG decidua (n = 5808); multiple other NEG mouse tissues[21] (n = 22,641); and embryonic small intestine from BEG mice[22] (n = 2685) identifies decidual-specific and common Blimp1 binding sites. **b** Over-represented gene categories associated with common (n = 1129) and decidual-specific Blimp1 ChIP peak subsets (n = 2962) within 100 kb of gene TSSs based on PANTHER pathway as analysed using GREAT V3.0 bionomial test with multiple testing correction. The number of genes associated with each gene category are indicated. **c** MA plot of E6.5 Blimp1 mutant vs wild-type decidual RNA-Seq profiles with up- (green) and down-(red) regulated E6.5 NEG Blimp1 target genes indicated. Candidate Blimp1 target genes with potential links to mutant decidual abnormalities are highlighted. **d** Over-represented gene categories among upregulated Blimp1 target genes based on GO Biological process with affinity propagation as analysed using Webgestalt 2019 with Benjamini-Hochberg multiple testing correction. The number of genes associated with each gene category are indicated.

decidualisation and identify candidate Blimp1 target genes, we exploited mice expressing an endogenous N-terminal EGFP-tagged Blimp1 protein (hereafter referred to as NEG) in combination with a proven GFP antibody for ChIP-Seq analysis[21]. We identified 5808 high-confidence genome-wide Blimp1 binding sites (Fig. 5a, Supplementary Data 5). Of these, 2846 overlapped with those previously identified in E18.5 small intestine and multiple diverse cell types[21,22]. Among shared ChIP peaks we identified 935 candidate Blimp1 target genes containing at least one peak within 100 kb of the transcriptional start site (TSS). This group included genes involved in JAK/STAT, Wnt and EGF-receptor signalling pathways (Fig. 5b, Supplementary Data 6). On the other hand, 2962 ChIP-Seq peaks associated with 1971 genes (Blimp1 ChIP-Seq peak ± 100 kb TSS) were unique to decidua (Fig. 5a). These included an over-representation of genes involved in gonadotrophin-releasing hormone receptor, integrin, angiogenesis and TGF-β signalling pathways (Fig. 5b, Supplementary Data 6).

Our comparison of Blimp1 ChIP-Seq peaks with differentially expressed transcripts identified approximately 26% of upregulated (n = 181) and 22% of downregulated genes (n = 102) in E6.5 Blimp1 mutant decidua as candidate Blimp1 targets. Several downregulated genes potentially contribute to the observed phenotypic abnormalities (Fig. 5c, Supplementary Data 7). For example, as mentioned above, expression of the metalloproteinase inhibitor Timp3 is markedly reduced in Blimp1 mutant decidua

at both E6.5 and E5.5. Additionally, *Nr2f2*, encoding COUP-TF2, is downregulated (2.02-fold) in mutant decidua. Conditional *Nr2f2* deletion in uterine tissues results in implantation and decidualisation defects[23]. *Dsp*, encoding desmoplakin, an essential cytoskeletal linker protein required for the assembly of functional desmosomes, is similarly downregulated (2.72-fold)[24]. Expression of desmin-containing intermediate filaments in endometrial stromal cells is thought to promote homophilic cell adhesion[25]. We speculate that decreased *Dsp* expression probably contributes to loose stromal cell contacts in Blimp1 mutant decidua.

In contrast, candidate Blimp1 targets with upregulated expression in E6.5 Blimp1 mutant decidua are enriched for genes associated with immune function and responses to other organisms (Fig. 5d, Supplementary Data 8). Previous experiments demonstrate that Blimp1 governs reprogramming of the post-natal intestinal epithelium and, in direct competition with the activator IRF-1, silences the MHC class I peptide-loading pathway to maintain tolerance during the suckling to weaning transition[22,26]. Consistent with its ability to repress expression of key components of the MHC class I peptide loading machinery in other tissues, we also found here that Blimp1 silences *Psmb8*, *Psmb9* and *Tap1* transcription.

**Upregulated Csf-1 expression and macrophage recruitment.** Our ChIP-Seq experiments demonstrate occupancy at the *Csf1*

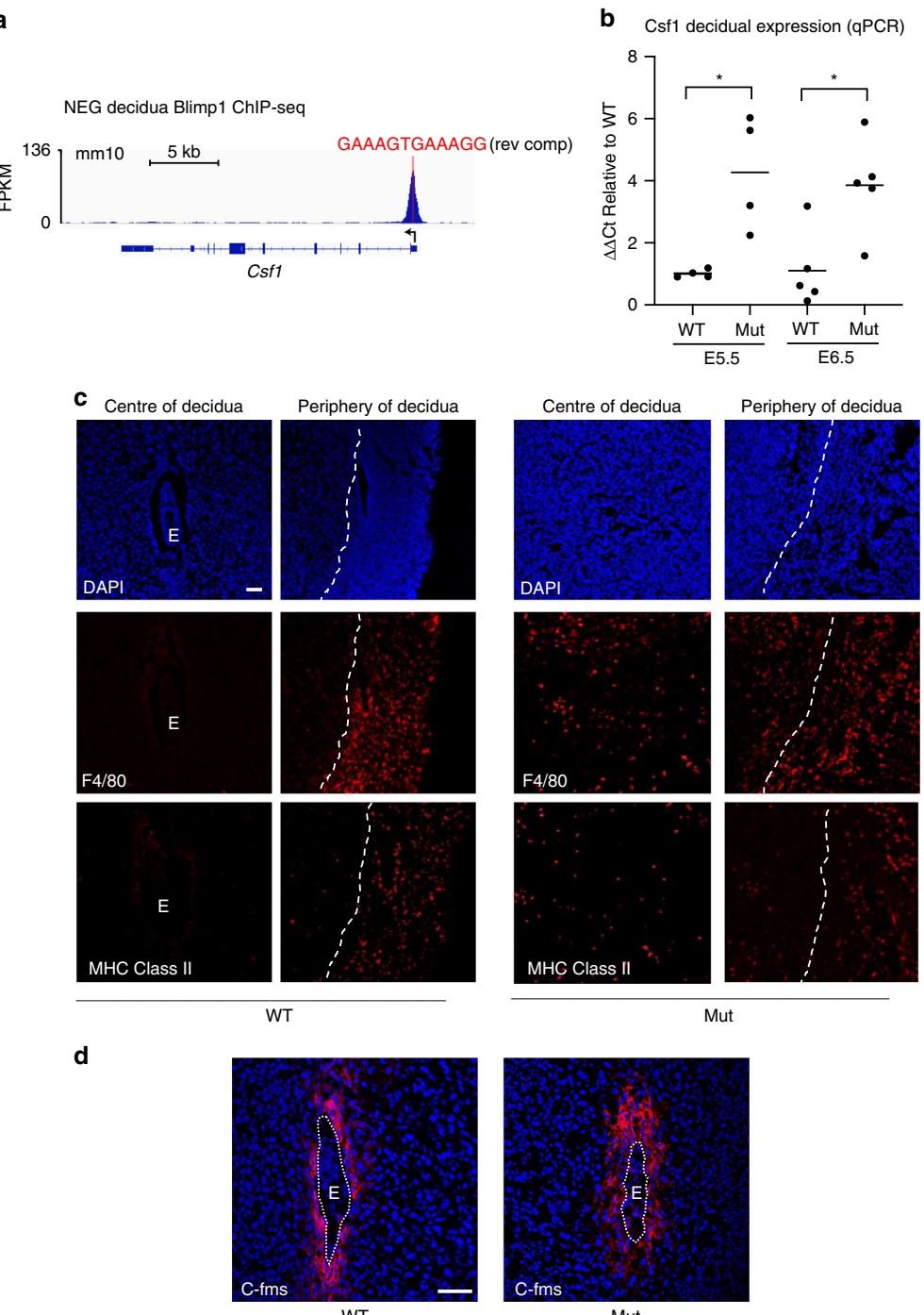

**Fig. 6 Blimp1 mutant decidua displays increased Csf1 and macrophage invasion into the PDZ. a** ChIP-Seq analysis identifies significant Blimp1 binding at the promoter of the *Csf1* gene in mouse decidua. A canonical Blimp1 binding motif underlies the ChIP-seq peak. **b** QPCR analysis confirms upregulated Csf1 expression in Blimp1 mutant decidua at both E5.5 and E6.5. Data from 4 (E5.5) and 5 (E6.5) samples per genotype. Bars represent mean. Two-tailed unpaired Student's *t*-test *$p = 0.0123$ for E5.5 and 0.0138 for E6.5. Source data are provided as a Source Data file. **c** IF staining of F4/80 and MHC Class II in E6.5 decidua with DAPI counterstain identifies infiltration of macrophages into central decidual regions in mutants. In contrast, staining in wild types is largely restricted to the surrounding myometrium ($n = 3$ decidua per genotype). White dotted lines mark the boundary between decidual stroma and myometrium. **d** IF staining of c-fms in E6.5 decidua with DAPI counterstain. C-fms is strongly expressed in the maternal decidua directly adjacent to the embryo in both mutant and wild types. Images are representative of triplicate staining experiments. E = embryo. Scale bars = 100 μm.

gene promoter in E6.5 mouse decidua (Fig. 6a). Comparison with previously published Blimp1 ChIP-Seq data sets likewise shows Blimp1 occupancy at the *Csf1* promoter is widely detectable in other mouse tissue types[21,22,27]. It has been suggested that highly conserved Blimp1 binding sites are largely non-functional[21]. However, here our qPCR analysis confirmed upregulated

expression of *Csf1* in Blimp1 mutant decidual tissues (Fig. 6b). Blimp1 was previously shown to bind the human Csf1 gene promoter[28]. Moreover, *CSF1* was among the most significantly upregulated genes in U266 cells following siRNA-mediated knockdown of Blimp1[28] confirming that occupancy of the human Csf1 promoter silences gene expression.

Wild-type decidua is mostly devoid of macrophages[29,30]. As expected, here we found that macrophage marker gene expression is largely restricted to the surrounding myometrium (Fig. 6c). However, upregulated Csf1 expression in Blimp1 mutant decidua was associated with macrophage invasion deep into decidual tissues (Fig. 6c, Supplementary Fig. 4) and upregulated expression of macrophage marker genes (Supplementary Fig. 2b). Moreover, F4/80, MHC Class II and CD74 immuno-reactive cells were present throughout Blimp1 mutant decidua (Fig. 6c, Supplementary Fig. 3a). Quantification of F4/80 and MHC Class II positive cells demonstrates a significant increase in the spread of macrophages (Supplementary Fig. 3b) as well as increased total macrophage numbers (Supplementary Fig. 3c). These results strongly suggest that Blimp1 normally silences Csf1 expression to prevent macrophage invasion into the implantation site.

C-fms/Csf1r, the only known Csf1 cell surface receptor, broadly expressed on cells of the myeloid mononuclear phagocytic lineage, mediates cell activation and chemotaxis[31–33]. Additionally, c-fms expression has been reported on trophoblasts and a restricted population of decidualised stromal cells at implantation sites[34–36]. Immunostaining confirmed abundant c-fms expression within the PDZ immediately adjacent to the embryo at E6.5 in both wild-type and mutant decidua (Fig. 6d). Expression was also observed on invading macrophages in Blimp1 mutant decidua (Supplementary Fig. 5). However, as expected c-fms was un-detectable on trophoblasts at these early stages of post-implantation development[35].

## Discussion

The present experiments demonstrate that the zinc finger transcriptional repressor Blimp1/PRDM1 is a critical regulator of the maternal decidual response during early pregnancy. Interestingly, the gene regulatory networks controlling functionality of the maternal decidual cell type have been shown to closely resemble those during inflammatory and cellular stress responses[37,38]. Similarly, Blimp expression is induced downstream of the unfolded protein response by diverse stress stimuli[39]. Moreover, our transcriptional profiling experiments demonstrate that expression of serum amyloid protein SAA3, an acute phase response protein implicated as a pro-inflammatory mediator[40,41], is robustly upregulated (272-fold) in mutant decidua. Collectively these observations strongly suggest that Blimp1 normally functions to silence maternal inflammatory responses during early post-implantation stages of pregnancy.

Here we describe Blimp1 expression patterns and essential functional activities within the maternal uterine environment during early stages of pregnancy. Blimp1 expression is robustly upregulated coincident with embryo attachment at E4.5 and maintained in the PDZ surrounding the embryo. To explore Blimp1 functional contributions within the maternal uterine tissues, we made use of the PR-Cre deleter strain[8] to selectively inactivate Blimp1 expression in the progesterone responsive decidualising stroma. Blimp1 functional activity is non-essential for uterine receptivity, blastocyst attachment and implantation. Moreover, decidualisation is correctly initiated at E4.5. However, the process of decidualisation arrests prematurely.

Our recent experiments demonstrate that Blimp1 silences expression of key components of the MHC class I peptide loading pathway and directly blocks IRF-1 occupancy at these sites to prevent premature activation of MHC class I surface expression and maintain neonatal tolerance in the developing intestine[22]. Similarly, here our ChIP-Seq analysis identified interferon-inducible components of the MHC class I peptide loading pathway as Blimp1 target genes normally silenced in the maternal decidual tissues. It has been known for many years that

Blimp1 silences IFNγ expression during Th1/Th2 T lymphocyte differentiation to shift the developmental program[42]. The present results demonstrate that Blimp1 plays a crucially important functional role dampening expression of the proinflammatory cytokine IFNγ and its downstream effectors in the context of the maternal uterine tissues at early post-implantation stages of pregnancy. Thus, IFNγ and many well-known IFNγ-responsive genes governing innate and adaptive immune responses were markedly upregulated in Blimp1 mutant decidua.

Early immunostaining and in situ hybridisation experiments suggested that uterine epithelial cells, natural killer (NK) cells, macrophages and placental trophoblasts at the maternal-fetal interface during mid-gestation all have the ability to produce IFNγ[43]. However, this signal was barely detectable at early post-implantation stages. Here we found in the absence of Blimp1 that upregulated IFNγ expression was associated with increased macrophage recruitment into the maternal decidual tissue. However, the numbers of uterine NK (uNK) cells remained constant (Supplementary Fig. 2c). Increased MHC class II staining similarly suggests that Blimp1-mediated repression of IFNγ production helps to maintain an immunologically privileged environment. However, additional studies will be required to clarify the connection between Blimp1 conditional loss in maternal cells expressing the progesterone receptor and those responsible for production of the inflammatory cytokine IFNγ.

Our RNA-Seq analysis demonstrates that expression of numerous genes involved in cell adhesion that normally accompany the decidual response was significantly downregulated in the Blimp1 mutants. Formation of the so-termed PDZ, the avascular region of tightly adherent decidual cells, physically constrains the embryo and prevents invasion of the embryonic trophoblasts into the maternal environment. Thus, a unique feature of Blimp1 mutant maternal decidual cells is their inability to correctly form the PDZ barrier. Formation of the PDZ is accompanied by robust induction of proteinase inhibitors. For example, expression of the metalloproteinase inhibitor Timp3 is normally induced in the PDZ at E6.5[17,44]. Here we found that Timp3 expression is markedly reduced in Blimp1 mutant decidua at both E5.5 and E6.5. Additionally, our ChIP-Seq analysis identified Timp3 as a candidate Blimp1 target gene suggesting that Blimp1 may also function as a transcriptional activator. Support for this idea also comes from recent experiments showing that Blimp1 functions as a transcriptional activator in the B-cell lineage[45]. Unlike other family members Timp3 is ECM bound and probably has only short-range localised activities[46]. Maternal Timp3 expressed immediately adjacent to the post-implantation embryo is thought to regulate trophoblast invasion into maternal tissues[44]. Mmps have also been implicated in trophoblast invasion[47] and Mmp11 is increased in Blimp1 mutant decidua. We speculate that imbalanced Timp3/Mmp11 activities potentially accounts for the increased trophoblast spreading and migration in Blimp1 mutant decidua.

Stromal cells within Blimp1 mutant decidua are loosely organised and have reduced tight junction-mediated barrier formation within the PDZ (Fig. 3f). Desmosomes in particular are implicated in tight junction formation in stromal cells during decidualisation[25]. Notably, expression of Dsp encoding desmoplakin, a key structural component required for functional desmosome formation, is reduced in Blimp1 mutants[24]. Dsp was also identified as a potential Blimp1 target gene. These findings suggest that activation of Dsp expression by Blimp1 promotes homophilic stromal cell interactions to maintain PDZ barrier function.

The activities of both BMP and Wnt ligands induced in the early uterine stroma surrounding the crypts are essential to promote the decidualisation process. Loss of either Bmp2 or the Bmp type 1 receptor Alk3 results in a complete failure of the decidualisation response[48,49]. Wnt signalling down-stream of

BMP activity stimulates decidual morphogenesis. The embryo mis-location phenotype observed here is reminiscent of that described for Wnt5a mutants[50]. Interestingly, comparison of Blimp1 ChIP-Seq peaks identified here in maternal decidual cells with those previously reported for multiple cell types[21,22] shows that the common peaks include components of BMP and Wnt signalling pathways. The present results strongly suggest that Blimp1 directly silences expression of the BMP and Wnt antagonists Grem1 and Sfrp1, respectively. Conversely, Blimp1 may also positively regulate transcription of genes upstream of BMP signalling. For example, Nr2f2/COUP-TFII, identified as a candidate Blimp1 target gene by ChIP-Seq, was downregulated in E6.5 mutant decidua. Additionally, we found that many of the peaks unique to decidua include regulators of gonadotrophin-releasing hormone receptors and TGF-β signalling. It will be interesting to learn more about the hierarchy of Blimp1-dependent transcriptional networks controlling decidualisation.

Growth factors produced by maternal decidual cells that regulate differentiation and migration of trophoblast cell populations is poorly understood. A unique feature of the Blimp1 mutant phenotype is the striking expansion of parietal trophoblast giant cells. Blimp1 expression in the developing placenta at mid-gestation stages has been previously described[6]. The present experiments strongly suggest that Fgf7 is a direct Blimp1 target gene. Upregulated Fgf expression could potentially lead to ectopic activation of the P-TGC subset. Addition of Fgf7 to cultured blastocysts results in increased P-TGC numbers and precocious differentiation[51]. Blimp1 probably dampens Fgf7 signalling to maintain the balance of TGC proliferation versus differentiation within the decidual stroma. Expression of Prl family members specific to P-TGCs is also markedly elevated. Increased numbers of P-TGCs probably reflect premature differentiation of the ExE trophoblast progenitor population. Likewise, expression of the Blimp1 target gene pregnancy associated plasma protein A (Pappa) is upregulated in the mutant. This metalloproteinase, a component of the IGF signalling pathway, regulates IGF bioavailability via cleavage of IGFBP4[52]. It seems likely that increased Pappa activity, leading to higher levels of IGF, promotes expansion of the primary TGC population[53].

RNA-Seq experiments demonstrate that upregulated transcripts in Blimp1 mutants are greatly enriched for genes related to immune cell function and chemotaxis. ChIP-Seq analysis identified many of these as candidate Blimp1 targets. The peak located at the Csf1 TSS previously reported in both human and mouse data sets is shared across multiple cell types[21,28]. The cytokine Csf1 controls macrophage recruitment into the cycling mouse uterus[54]. High doses of Csf1 have been shown to induce embryo resorption[55]. Similarly, here we found that upregulated Csf1 expression is associated with increased macrophage invasion into the maternal decidual tissues, loss of the PDZ barrier function and ectopic expansion of fetal extra-embryonic trophoblast cell populations. Additional experiments will be necessary to dissect the contributions made by individual Blimp1 target genes that are collectively responsible for these tissue disturbances.

## Methods

**Animal care and use.** Female C57BL/6 mice (6–10 weeks of age) were used as the wild-type strain. $Prdm1^{BEH/+};PR^{Cre/+}$ males were generated by crossing $Prdm1^{BEH/+}$ mice[5] with $PR^{Cre/+}$ mice[8]. To induce Prdm1 gene deletion in the uterus, these males were crossed with female $Prdm1^{CA/CA}$ mice[9] to generate $Prdm1^{BEH/CA};PR^{Cre}$ females (referred to as Blimp1 mutants throughout the text). In some experiments Blimp1 mutant females were mated with $Rosa26^{mT/mG}$ males[12] for immuno-fluorescence analysis. For BrdU labelling experiments, 0.25 mg of 5-bromo-2-deoxyuridine (BrdU, Sigma) per gram of body weight in phosphate buffered saline (PBS) was injected intraperitoneally 2 h prior to sacrifice. Genotyping was performed as described in the original reports. All animal experiments were performed in accordance with the UK Home Office regulations and approved by the University of Oxford Local Ethical Committee.

**Chicago Blue visualisation and quantification.** To visualise implantation sites, mice were injected intravenously with 0.1 ml of 1% (w/v) Chicago Sky Blue 6B (Sigma) in PBS under anaesthetic and sacrificed by $CO_2$ asphyxiation 10 min later. Individual decidua were removed from the uterus and shaken in 25 μl of for-mamide for 2 days at room temperature (RT), protected from light to extract the dye. After clearing by centrifugation (16,000 × g for 5 min), dye content in for-mamide extracts was quantitated spectrophotometrically using an ND1000 Nanodrop at 618 nm relative to a standard curve of Chicago Sky Blue dye dissolved in formamide.

**Western blot analysis.** RIPA lysates (30 μg per sample) were analysed by Western blot as previously described[56]. Antibodies used are listed in the Supplementary Table 1. Uncropped blots can be found in the Source data file.

**Immunohistochemistry.** For immunohistochemistry (IHC), virgin and pregnant (E3.5, E4.5, E5.5 and E6.5) uteri or individual decidua were fixed overnight in 4% paraformaldehyde (PFA) in PBS, dehydrated using an ethanol series, embedded in paraffin wax and sectioned (6 μm). Dewaxed sections were subjected to antigen retrieval by boiling for 1 h in Tris/EDTA (pH 9.0) and permeabilised for 10 min in 0.1% Triton X-100 (Sigma) in PBS. After blocking with 10% normal goat serum in PBS with 0.05% Tween-20 (Sigma) for 1 h at RT, sections were incubated with primary antibodies in blocking solution overnight at 4 °C. Rat monoclonal antibodies underwent signal amplification with rabbit anti-rat secondary antibody (AI-4001, Vector Laboratories) for 45 min at RT. All samples were then subjected to peroxidase blocking for 20 min at RT and developed with Envison System-HRP for rabbit antibodies (K4011, DAKO) and Vector Red substrate (SK-4805, Vector Laboratories). Sections were lightly counterstained with haematoxylin, coverslipped and imaged. Haematoxylin and eosin staining was performed as per standard protocols. Antibodies used are listed in the Supplementary Table 1.

**Alkaline phosphatase staining.** Decidua (E5.5) from Blimp1 mutant and wild-type females were fixed overnight in 1% PFA in PBS at 4 °C, washed in PBS, cryoprotected using a sucrose gradient and embedded in OCT. Cryosections (8 μm) were cut using a Leica CM3050 S Research Cryostat, post fixed in 0.2% glutar-aldehyde in PBS for 15 min at RT, washed with PBS and stained with BCIP/NBT (11697471001, Sigma) according to the manufacturer's protocol.

**Immunofluorescence.** Freshly dissected whole decidua and isolated embryos were fixed in 1% PFA in PBS overnight at 4 °C followed by washing in PBS. For vibratome processing decidua were embedded in 4% agarose in PBS, manually trimmed and 100 μm sections cut using a Leica VT1000 S vibrating blade micro-tome. For cryosectioning, decidua were cryoprotected using a sucrose gradient and embedded in OCT before being sectioned at 8 μm on a Leica CM3050 S Research Cryostat.

Decidual sections or whole embryos were washed three times in PBS containing 0.1% Triton X-100 (PBS-T). Samples were permeabilised in PBS containing 0.5% Triton X-100 for 20 min, washed in PBS-T and then blocked in 5% donkey serum plus 0.2% BSA in PBS-T for 1 h at RT. Samples were incubated overnight with primary antibodies in blocking solution at 4 °C. After washing, samples were incubated with fluorophore-conjugated secondary antibodies in blocking solution for 2 h at RT. Following three subsequent washes in PBS-T, samples were washed in PBS-T containing 2 μg/ml DAPI for 15 minutes at RT, then washed three more times in PBS-T and mounted in Vectashield with DAPI (H-1200, Vector Laboratories). All samples were imaged on an Olympus Fluoview FV1000 confocal microscope and image data was processed and analysed using ImageJ. Antibodies are listed in Supplementary Table 1.

**Transmission electron microscopy.** For TEM analysis, E6.5 wild-type and Blimp1 mutant decidua were fixed with 4% PFA plus 2.5% glutaraldehyde in 0.1 M sodium cacodylate buffer at pH 7.2 for 4 h at RT then overnight at 4 °C. After embedding in 4% agarose in PBS, samples were manually trimmed and thick sections (200 μm) cut using a Leica VT1000 S vibrating blade microtome. The thick sections were processed using a Leica AMW microwave tissue processor, followed by infiltration with TAAB TLV resin over 3 days, and polymerisation for 48 h at 60 °C. Ultrathin (90 nm) sections were cut using a Diatome diamond knife on a Leica Ultracut7 ultramicrotome, post-stained with lead citrate for 5 min, and examined on a Tecnai 12 transmission electron microscope (FEI) equipped with a Gatan OneView CMOS camera.

**RNA-Seq.** Total RNA was extracted from E5.5 and E6.5 decidua from Blimp1 mutant and wild-type littermates ($Prdm1^{CA/+}$) using an RNeasy Mini kit (Qiagen Cat#74104) with on column DNase treatment according to the manufacturer's protocol. RNA was normalized to 800 ng per sample followed by depletion of cytoplasmic and mitochondrial ribosomal RNA sequences (Ribo-Zero Gold rRNA Removal Kit (H/M/R), Cat: #MRZG12324). Subsequent library preparation was performed using the Illumina TruSeq Stranded Total RNA Library Prep kit (H/M/R) (Cat: #20020597), followed by sequencing on an Illumina HiSeq4000 (75 bp paired end).

**RNA-Seq data analysis**. Paired-end sequencing reads (75 bp) from mouse decidual samples were mapped to the mm10 mouse genome using RNA-STAR in Galaxy (https://usegalaxy.org). Aligned BAM files were then analysed using the RNA-Seq quantitation pipeline in SeqMonk (V1.45.4). Differentially expressed genes were identified using DeSeq2 with a FDR cutoff of 0.05, >2-fold change in expression and FPKM of ≥1 (in all samples within at least one group). Data was visualised using Seqmonk, PCA, MA plot and datastore tree (hierarchal clustering) functions. GO analysis was performed using WebGestalt (www.webgestalt.org)[57]. Gene set enrichment analysis (GSEA) was performed using all RNA-Seq genes ranked by DESeq2 FDR from most significantly upregulated to most significantly downregulated and compared with genes displaying PR binding 1 hour following P4 treatment[58] (GSE40663) or Esr1 binding 1 h following E2 treatment[59] (GSE36455) in uterine tissues. PR and Esr1 ChIP-Seq regions were converted from mm9 to mm10 using LiftOver[60] and gene/peak associations identified using GREAT[61] based the single nearest gene option ± 10 kb of gene TSSs. IFNγ responsive genes (GO:0034341, filter: Mus musculus) were identified using AMIGO2 (http://amigo.geneontology.org/amigo)[62].

**ChIP-Seq analysis**. To identify genome-wide Blimp1 binding sites, triplicate pools (n = 6–8) of freshly dissected decidua (E6.5) from homozygous NEG mice expressing an endogenous N-terminal EGFP-tagged Blimp1 protein[21] were minced using a razor blade in 50 µl of culture media (RPMI-1640, 10% FCS, 0.0004% β-mercaptoethanol, 1x Pen/Strep, 2 mM glutamine) and cross-linked in 1% formaldehyde in culture media for 20 min at RT. Pools of decidua from wild-type mice were processed in parallel as a negative control. After quenching in 0.125 M glycine for 5 min, samples were washed in PBS and processed for ChIP using 6 µg of rabbit anti-GFP antibody (Ab290, Abcam). Triplicate test (GFP ChIP of NEG) and control (GFP ChIP of wild type) decidual samples, and their respective input samples were analysed. Sequence reads were mapped to the mm10 genome using bwa-MEM (0.7.15-r1140). After removing PCR duplicate pairs using SAMTools-0.1.19[63], peak calling was performed with MACS2[64] using default parameters to call areas of enrichment in ChIP samples over input. High-confidence peaks were identified by intersecting peak areas in the triplicate GFP ChIP of NEG samples. Non-specific peaks were removed by subtracting overlapping peaks called in triplicate GFP ChIP of wild-type samples.

For comparison with NEG Blimp1 ChIP-Seq data, mouse embryonic small intestine Blimp1 ChIP-Seq data[22] (GSE66069) was converted from mm9 to mm10 using LiftOver[60]. Blimp1 ChIP peaks in multiple other NEG mouse tissues were previously reported[21]. Gene/peak associations were identified using GREAT[61] based the single nearest gene option ± 100 kb of gene TSSs.

**QPCR**. RNA (1 µg) was reverse transcribed to cDNA using Superscript III First Strand Synthesis System (Life Technologies, Cat#18080-051) and oligo dT. QPCR was performed with QuantiTect SYBR Green PCR mix (Qiagen) using 50 ng of cDNA per reaction. Relative gene expression was calculated using the ddCT method using Hprt as the housekeeping gene. QPCR primer sequences are listed in Supplementary Table 2.

**Statistical analysis**. Experiments were repeated at least three times. Exact n is stated for every experiment in figure legends. GraphPad Prism 8.0 software was used for statistical analysis. Student's unpaired two tailed t-tests were used for statistical analyses. $P < 0.05$ was considered statistically significant. Sample sizes were selected based on current and previous experiments and no statistical method was applied to predetermine sample size. Experiments were not randomized and investigators were not blinded during experimentation or assessment.

## Data availability

RNA-Seq and ChIP-Seq data have been deposited in the NCBI GEO database under accession code: GSE141613. All other relevant data is available from the authors.

Publicly available source data in Figs. 4 and 5 were obtained from NCBI GEO under accession codes: GSE40663, GSE36455, GSE66069, and GSE91038.

The source data underlying Figs. 1b–c, 2b, e, 3e, 4f, 6b and Supplementary Figs 1a, 2c, and 3b–c are provided as a Source Data file.

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

## Acknowledgements

We would like to thank Satoko Nishimoto for help with pilot experiments as well as Errin Johnson and Raman Dhaliwal for assistance with TEM experiments. We thank the Oxford Genomics Centre at the Wellcome Centre for Human Genetics (funded by Wellcome Trust grant reference 203141/Z/16/Z) for the generation and initial processing of the sequencing data. We thank Mitinori Saitou for supplying the *Prdm1*EGFP (NEG) targeted ES cells. Confocal microscopy was carried out in the Micron Advanced Bioimaging Unit (funded by Wellcome Trust Strategic Award 107457). Generation of the Pgr-cre driver was funded by the NIH/ NICHD (R01HD042311 to J.P.L.). This work was funded by the Wellcome Trust (214175/Z/18/Z to E.J.R.). E.J.R. is a Wellcome Trust Principal Research Fellow.

## Author contributions

M.G., E.K.B., E.J.R. and A.W.M. designed the experiments. M.G., M.-E.X., I.C. and A.W.M. performed the experiments. M.G., E.K.B., E.J.R. and A.W.M. contributed to writing the paper. J.P.L. and F.J.D. provided the *PR*^Cre/+^ mice.

## Competing interests

The authors declare no competing interests.
