## [Peer Review File · Nature Communications]

Reviewers' Comments:

Reviewer #1:

Remarks to the Author:

Decidualisation of uterine tissue plays an essential role in regulating the invading trophoblast tissue. Moreover, the blastocyst implantation site is immunologically privileged. This carefully conducted study reports on the role of BLIMP1, a widely expressed transcriptional repressor, in some of the uterine cells during decidualisation at the onset of blastocyst implantation. Loss of BLIMP1 results in a more significant invasion of uterine tissue by trophoblast cells, up-regulation of immune control gene IFN γ , and Csf-1 and macrophage recruitment at the site, and the loss of the appropriate maternal immune response to invading trophoblast cells. Consequently, the development of the embryo ceases at the early stages of development. The Figures complement the text, and together they show the consequences of lack of BLIMP1 during implantation. The results of the study are presented clearly and do not have any significant flaws.

The authors acknowledge that there are some inevitable gaps, for example, it is unknown how precisely BLIMP1 represses the genes during early decidualisation. Based on previous examples, they speculate that an unknown histone modifier might be involved in conjunction with BLIMP1 for the repression.

Some specific points are as follows:

1. Fig 1A: In uteri at E3.5 post fertilisation, there is a marked increase in BLIMP1 expression in the stroma. What triggers the upregulation before implantation? How does this change compare with any cyclical changes during the estrus cycle? An increase in BLIMP1 expression is evident at the site of blastocyst implantation at E5.5 onwards. What causes this increase, a specific signal from the implanting blastocyst, or due to mechanical forces?
2. The number of decidual cells in mutant and wt decidual cells was unaffected, but what were the actual numbers. Fig 2B. Is it possible to give the values for the levels of Chicago Blue detected at the deciduae in regular and BLIMP1 mutant sites?
3. The absolute levels of IFN γ (Fig 4 E) was very low despite its proposed increase in BLIMP1 KO. Some comment on this and its potential effect on the phenotype would be helpful, perhaps by comparison with other examples?
4. Fig 6B and 6D. The difference in Csf1 in wt and mutants: Is this also due to changes in the number of cells in the decidua compared to controls?
5. Fig 1D. Additional labeling of the figure would help readers not familiar with the implantation site.

Finally, have the authors considered if some genes might be induced by BLIMP1?

Reviewer #2:

Remarks to the Author:

This paper examines the requirement for Blimp1 in early pregnancy using a conditional depletion of it in uterine tissue through the use of a PR-Cre. The author's convincingly show that the early decidual response in the uterus is impaired with resultant elevated invasion of trophoblasts and loss of embryonic viability. Mechanistically the authors use RNA seq coupled to expression analysis using in situ methods to suggest possible roles for Blimp1. One focus was upon CSF1, a central regulator of macrophage biology, whose expression is aberrantly upregulated upon loss of Blimp1 and the consequence of macrophages found in the decidual tissue in the mutant.

Overall the study is well done with a considerable amount of data. The major problem is that Blimp1 clearly regulate many different genes in many different cells and thus the deduction of causality is difficult. There were statements about up-regulation of proteases that enhance trophoblast invasion but no data to support this. This is true for all other genes including INFg.

There was significant discussion about Csf1 that looks to be a direct target for Blimp1. Thus, Blimp1 loss resulted in an up-regulation of CSF1 that correlated with increased invasion of macrophages into the decidua from where they are usually excluded. A claim was made that this and deregulation of inflammatory cytokines resulted in breakage of immune tolerance and thus loss of embryos. However, as far as I could see, these were not allogeneic pregnancies (not clear as strains were not specified but should be) thus there should be no anti-fetal immune response. Furthermore, although loss of Csf1 does impair fertility but this is independent on it being in an inbred or out-bred pregnancy. Thus, this conjecture seems misplaced and emphasizes the difficulty of deducing causality from expression data. Could these macrophages be in the decidua to eat dying decidual cells?

Given the focus on Csf1 it would be important to show that Blimp1 was ablated in the uterine epithelium as this is the major source during early pregnancy. In addition it would be important to show elevated expression in the epithelium or whether there is promiscuous expression in other sites due to the normal repressive role of Blimp1.

Minor problems:

1. Efficiency of PR-cre is shown only by IHC and this is very hard to determine from the figure as the background is high. It need quantitation, is it variable in the different tissues?
2. The reference for Csf1 inducing macrophage chemotaxis (Sasmono et al) is incorrect. The first description was Webb et al 1996.
- 3, References for macrophage exclusion from the decidua should be included, Tachi and Tachi 1981 first in rats, Pollard et al 1991 in mice.
3. Csf1r expression in trophoblast and a detailed expression profile of it and Csf1 was published in Arceci et al 1989.
4. It is a bit surprising that the Csf1r expression does not appear to be on macrophages in the decidua that from the previous data should be in the mutant in this area (Fig 6D). This does bring into doubt the validity/sensitivity of the antibody.

Reviewer #3:

Remarks to the Author:

This is a really exciting paper, which identifies an important regulator of endometrial development in pregnancy. The experiments comprehensively address their research question, and the manuscript is well written. I have a few comments that I think should be considered.

The authors identify that regulation of Blimp1 appears to have a substantial impact on inflammatory regulatory pathways. This is explained away as potentially supporting maternal-fetal tolerance. However, the authors do not consider that the decidual cell type evolved as a result of the inflammatory consequences of maternal fetal interactions. I think this concept needs to be introduced into the manuscript. It is very significant that an inflammatory gene regulator might be acutely important for facilitating placentation.

Minor comments

- In a few instances the authors simply refer to 'up regulated genes' and 'down regulated genes' I think that it would be much easier to follow the manuscript if the authors specify the particular comparison being discussed. For example, the first sentence in the last paragraph of the discussion.
- Can the authors explain the difference between their hierarchical clustering and PCA. The authors use the hierarchical clustering to argue that E6.5 of the mutants is embedded within the E5.5 of the wildtype, however, the PCA analysis suggests that each E6.5 cluster is equally diverged from the E5.5 animals. I suspect the observed hierarchical clustering pattern is simply an artefact of trying to reduce the data down to a single dimension.

We were pleased to receive strongly positive comments from all three Reviewers:

Reviewer 1: "This carefully conducted study reports on the role of Blimp1.... during decidualization at the onset of blastocyst implantation... The results of the study are presented clearly and do not have any significant flaws."

Reviewer 2: "Overall the study is well done with considerable amount of data."

Reviewer 3: "This is a really exciting paper.... The experiments comprehensively address their research question, and the manuscript is well written."

As detailed below, we have revised the text and modified several figures in response to the helpful and constructive comments made by the reviewers.

Referee #1: Decidualisation of uterine tissue plays an essential role in regulating the invading trophoblast tissue. Moreover, the blastocyst implantation site is immunologically privileged.... Loss of BLIMP1 results in a more significant invasion of uterine tissue by trophoblast cells, up-regulation of immune control gene IFN gamma, and Csf-1 and macrophage recruitment at the site, and the loss of the appropriate maternal immune response to invading trophoblast cells.... The Figures complement the text, and together they show the consequences of lack of BLIMP1 during implantation.... The authors acknowledge that there are some inevitable gaps, for example, it is unknown how precisely BLIMP1 represses the genes during early decidualisation. Based on previous examples, they speculate that an unknown histone modifier might be involved in conjunction with BLIMP1 for the repression.

Some specific points are as follows:

1. Fig 1A: In uteri at E3.5 post fertilisation, there is a marked increase in BLIMP1 expression in the stroma. What triggers the upregulation before implantation? How does this change compare with any cyclical changes during the estrus cycle? An increase in BLIMP1 expression is evident at the site of blastocyst implantation at E5.5 onwards. What causes this increase, a specific signal from the implanting blastocyst, or due to mechanical forces?

NFκ-B signals have been shown to activate Blimp1 expression during plasma cell terminal differentiation (Morgan et al., 2009), whereas BMP/SMAD signals function as upstream regulators in the PGC cell lineage (Lawson et al., 1999; Tremblay et al., 2001). We agree that learning more about growth factor signalling and hormonal regulation of Blimp1 expression during the oestrus cycle and pregnancy is an important priority. It is possible that mechanical forces could also potentially play a role. These interesting issues will be examined in future studies.

2. The number of decidua in mutant and wt decidua was unaffected, but what were the actual numbers. Fig 2B.

Further detailed information has now been inserted into the 2nd paragraph on page 5 of the manuscript as follows: “The average number of decidua/mouse \pm SEM was 9.88 ± 0.42 for wild types (n=25) and 9.38 ± 0.38 for mutants (n=24)”.

Is it possible to give the values for the levels of Chicago Blue detected at the deciduae in regular and BLIMP1 mutant sites?

Values for Chicago Blue levels have now been inserted into the legend of Fig. 2 as follows: “Chicago Blue content is reduced in decidua but not the ovaries of mutants (0.271 ± 0.013 μ g/decidua) in comparison to wild type (0.348 ± 0.008 μ g/decidua) mice”.

3. The absolute levels of IFN gamma (Fig4 E) was very low despite it's proposed increase in BLIMP1 KO. Some comment on this and its potential effect on the phenotype would be helpful, perhaps by comparison with other examples?

Yes, we agree that *Ifn γ* expression levels detectable by RNA sequencing were very low. However, we used the more sensitive method of qPCR to demonstrate significantly increased expression levels. These results strongly suggest that *Blimp1* silences expression of *Ifn γ* and downstream target genes to maintain an immunoprivileged environment within maternal uterine tissues.

4. Fig 6B and 6D. The difference in Csf1 in wt and mutants: Is this also due to changes in the number of cells in the decidua compared to controls.

This seems unlikely. Thus, IHC staining at both E5.5 and E6.5 identified cytoplasmic *Csf1*-expressing decidual cells immediately surrounding the implanted embryo. Increased staining was detected in the same cell population in mutant decidua (Supplementary Fig. 4).

5. Fig 1D. Additional labeling of the figure would help readers not familiar with the implantation site.

As suggested by the Reviewer, additional labels have been added to Figure 1 to identify key features of the implantation site.

Finally, have the authors considered if some genes might be induced by BLIMP1?

Yes, besides *Timp3*, we now comment on additional genes induced by *Blimp1* on pages 9 and 13 of the revised manuscript. Specifically, *Dsp* (Desmoplakin) is down-regulated 2.72-fold in E6.5 mutants and has a *Blimp1* ChIP peak 27.5 kb upstream of the TSS. Interestingly, *Dsp*

is known to be important for desmosome related stromal cell–cell contacts during decidualisation. Nr2f2 (COUP-TFII) is down-regulated 2.02-fold in E6.5 mutants and has a Blimp1 ChIP peak 0.25 kb upstream of the TSS. Interestingly, Nr2f2 has been shown to function upstream of maternal BMP signalling during implantation. Loss of function mutants display defective blastocyst implantation and decidualisation. Together with similar observations reported by Minnich et al. (2016) in the B-cell lineage these findings suggest that some genes are induced by Blimp1.

Referee #2: *This paper examines the requirement for Blimp1 in early pregnancy using a conditional depletion of it in uterine tissue through the use of a PR-Cre. The author's convincingly show that the early decidual response in the uterus is impaired with resultant elevated invasion of trophoblasts and loss of embryonic viability. Mechanistically the authors use RNA seq coupled to expression analysis using in situ methods to suggest possible roles for Blimp1. One focus was upon CSF1, a central regulator of macrophage biology....The major problem is that Blimp1 clearly regulate many different genes in many different cells and thus the deduction of causality is difficult. There was significant discussion about Csf1 that looks to be a direct target for Blimp1. Thus, Blimp1 loss resulted in an up-regulation of CSF1 that correlated with increased invasion of macrophages into the decidua from where they are usually excluded. A claim was made that this and deregulation of inflammatory cytokines resulted in breakage of immune tolerance and thus loss of embryos. However, as far as I could see, these were not allogeneic pregnancies (not clear as strains were not specified but should be) thus there should be no anti-fetal immune response.*

Mutant strains used in the present study were not backcrossed. Rather, matings were performed on a 129xC57BL6 mixed genetic background using animals that were segregating coat colour markers and numerous minor histocompatibility loci. Up-regulated expression of the potent inflammatory cytokine $\text{Ifn}\gamma$ together with recruitment of macrophages into maternal decidual tissue seem likely to reflect maternal immune reactivity.

Furthermore, although loss of Csf1 does impair fertility but this is independent on it being in an inbred or out-bred pregnancy. Thus, this conjecture seems misplaced and emphasizes the difficulty of deducing causality from expression data.

Unlike defective fertility caused by the loss of Csf1, the mutant phenotype we describe here shows increased Csf1 expression associated with macrophage invasion into decidual tissues. Similarly, treatment with Csf1 has previously been shown to compromise fetal survival (Tartakovsky, 1989). Taken together, these observations strongly argue for a causal relationship.

Could these macrophages be in the decidua to eat dying decidual cells?

Possibly, however we observe no evidence suggesting the decidual cells are dying.

Given the focus on Csf1 it would be important to show that Blimp1 was ablated in the uterine epithelium as this is the major source during early pregnancy. In addition it would be important to show elevated expression in the epithelium or whether there is promiscuous expression in other sites due to the normal repressive role of Blimp1.

Blimp1 is selectively expressed in luminal epithelial cells at E4.5 implantation sites. At later stages (i.e. E5.5 and E6.5), when Csf1 expression increases in the mutants, these epithelial cells have all but disappeared from the implantation site. As mentioned above in response to reviewer 1, IHC staining in Blimp1 mutants at both E5.5 and E6.5 identified increased cytoplasmic Csf1 staining in decidual cells immediately surrounding the embryo (Supplementary Fig. 4). These observations strongly suggest that Blimp1 normally represses decidual Csf1 expression.

Minor problems:

1. Efficiency of PR-cre is shown only by IHC and this is very hard to determine from the figure as the background is high. It need quantitation, is it variable in the different tissues?

We agree that the quality of the original images was sub-optimal. These experiments have been carefully redone and new images with reduced background staining have been inserted. Additionally, we have quantified the area of Blimp1 staining on multiple sections of wild type and Blimp1 mutant decidua. A bar graph summary of the data has now been inserted in Figure 1C. Collectively, these results clearly demonstrate that the PR-Cre-driven deletion eliminates Blimp1 expression in decidualised stroma.

2. The reference for Csf1 inducing macrophage chemotaxis (Sasmono et al) is incorrect. The first description was Webb et al 1996.

3, References for macrophage exclusion from the decidual should be included, Tachi and Tachi 1981 first in rats, Pollard et al 1991 in mice.

3. Csf1r expression in trophoblast and a detailed expression profile of it and Csf1 was published in Arceci et al 1989.

As suggested by the reviewer, the correct references have now been inserted.

4. It is a bit surprising that the Csf1r expression does not appear to be on macrophages in the decidua that from the previous data should be in the mutant in this area (Fig 6D). This does bring into doubt the validity/sensitivity of the antibody.

Our IF images demonstrate Csf1r/c-fms expression in the entire PDZ region. IHC analysis (now included as Supplementary Fig. 5) shows c-fms staining on macrophages located around the implanted embryo in E6.5 Blimp1 mutant decidua. IHC staining of serial sections identifies F4/80 positive macrophages in the same region. The antibodies used for detection of c-fms and F4/80 are both rat monoclonal antibodies precluding the ability to perform double labelling experiments.

Reviewer #3 (Remarks to the Author): *The authors identify that regulation of Blimp1 appears to have a substantial impact on inflammatory regulatory pathways. This is explained away as potentially supporting maternal-fetal tolerance. However, the authors do not consider that the decidual cell type evolved as a result of the inflammatory consequences of maternal fetal interactions. I think this concept needs to be introduced into the manuscript. It is very significant that an inflammatory gene regulator might be acutely important for facilitating placentation.*

This important concept is now introduced at the start of the Discussion to replace the original paragraph containing largely repetitive background information presented in the Introduction, as follows:

“The present experiments demonstrate that the zinc finger transcriptional repressor Blimp1/PRDM1 is a critical regulator of the maternal decidual response during early pregnancy. Interestingly, the gene regulatory networks controlling functionality of the maternal decidual cell type have been shown to closely resemble those during inflammatory and cellular stress responses^{37,38}. The finding that Blimp expression is induced downstream of the unfolded protein response by diverse stress stimuli provides additional support for this way of thinking³⁹. Moreover, our transcriptional profiling experiments demonstrate that expression of serum amyloid protein SAA3, an acute phase response protein implicated as a pro-inflammatory mediator^{40,41}, is robustly upregulated (272-fold) in mutant decidua. Collectively these observations strongly suggest that Blimp1 normally functions to silence maternal inflammatory responses during early post-implantation stages of pregnancy.”

Minor comments

- *In a few instances the authors simply refer to ‘up regulated genes’ and ‘down regulated genes’ I think that it would be much easier to follow the manuscript if the authors specify the particular comparison being discussed. For example, the first sentence in the last paragraph of the discussion.*

As suggested by the reviewer this sentence has been modified to “RNA-Seq experiments demonstrate that up-regulated transcripts in Blimp1 mutants are greatly enriched for genes

related to immune cell function and chemotaxis.”

- *Can the authors explain the difference between their hierarchical clustering and PCA. The authors use the hierarchical clustering to argue that E6.5 of the mutants is embedded within the E5.5 of the wildtype, however, the PCA analysis suggests that each E6.5 cluster is equally diverged from the E5.5 animals. I suspect the observed hierarchical clustering pattern is simply an artefact of trying to reduce the data down to a single dimension.*

These analyses answer different questions. The hierarchical clustering (SeqMonk datastore tree, Supplementary Fig. 2A) is based on all genes (n=34, 235) and provides a one-dimensional view of global transcriptional similarities between samples. In contrast the PCA emphasises sample variation, identifying principal components based on gene subsets that separate samples. In this case we consider the PCA to be more informative. This analysis has been included in the main figure, whereas the hierarchical clustering has been supplied as supplementary data.

Reviewers' Comments:

Reviewer #2:

Remarks to the Author:

The authors have largely answered the comments of the original reviewers. More detailed IHC has been provided for macrophages, CSF1 and CSF1R. This data improved the manuscript. Further the author's have acknowledged the difficulty of ascribing causality given the multiple genes regulated by Blimp-1, several of which could have independent roles in the loss of embryo viability. The paper is well done and I think should now be published.

Point-by-point Response to Reviewers Comments

REVIEWERS' COMMENTS:

Reviewer #2 (Remarks to the Author):

The authors have largely answered the comments of the original reviewers. More detailed IHC has been provided for macrophages, CSF1 and CSF1R. This data improved the manuscript. Further the author's have acknowledged the difficulty of ascribing causality given the multiple genes regulated by Blimp-1, several of which could have independent roles in the loss of embryo viability. The paper is well done and I think should now be published.

We thank the reviewer for their comments and are happy they now this the paper is appropriate for publication.